# Potential Association of Cytochrome P450 Copy Number Alteration in Tumour with Chemotherapy Resistance in Lung Adenocarcinoma Patients

**DOI:** 10.3390/ijms241713380

**Published:** 2023-08-29

**Authors:** Evelyn Incze, Katalin Mangó, Ferenc Fekete, Ádám Ferenc Kiss, Ádám Póti, Tünde Harkó, Judit Moldvay, Dávid Szüts, Katalin Monostory

**Affiliations:** 1Institute of Enzymology, Research Centre for Natural Sciences, Magyar Tudósok 2, H-1117 Budapest, Hungary; gabri.evelyn@ttk.hu (E.I.); mango.katalin@ttk.hu (K.M.); fekete.ferenc@ttk.hu (F.F.); kissadamferenc@gmail.hu (Á.F.K.); poti.adam@ttk.hu (Á.P.); szuts.david@ttk.hu (D.S.); 2Doctoral School of Pharmaceutical Sciences, Semmelweis University, Üllői 26, H-1085 Budapest, Hungary; 3Department of Pathology, National Korányi Institute of Pulmonology, Pihenő 1, H-1121 Budapest, Hungary; harko.tunde@koranyi.hu; 41st Department of Pulmonology, National Korányi Institute of Pulmonology, Pihenő 1, H-1121 Budapest, Hungary; drmoldvay@hotmail.com

**Keywords:** copy number alteration, lung adenocarcinoma, CYP2C8, CYP3A4, paclitaxel-metabolizing capacity, resistance to anticancer therapy

## Abstract

Resistance to anticancer agents is a major obstacle to efficacious tumour therapy and responsible for high cancer-related mortality rates. Some resistance mechanisms are associated with pharmacokinetic variability in anticancer drug exposure due to genetic polymorphisms of drug-metabolizing cytochrome P450 (CYP) enzymes, whereas variations in tumoural metabolism as a consequence of *CYP* copy number alterations are assumed to contribute to the selection of resistant cells. A high-throughput quantitative polymerase chain reaction (qPCR)-based method was developed for detection of *CYP* copy number alterations in tumours, and a scoring system improved the identification of inappropriate reference genes that underwent deletion/multiplication in tumours. The copy numbers of both the target (*CYP2C8*, *CYP3A4*) and the reference genes (*ALB*, *B2M*, *BCKDHA*, *F5*, *CD36*, *MPO*, *TBP*, *RPPH1*) established in primary lung adenocarcinoma by the qPCR-based method were congruent with those determined by next-generation sequencing (for 10 genes, slope = 0.9498, r^2^ = 0.72). In treatment naïve adenocarcinoma samples, the copy number multiplication of paclitaxel-metabolizing CYP2C8 and/or CYP3A4 was more prevalent in non-responder patients with progressive disease/exit than in responders with complete remission. The high-throughput qPCR-based method can become an alternative approach to next-generation sequencing in routine clinical practice, and identification of altered *CYP* copy numbers may provide a promising biomarker for therapy-resistant tumours.

## 1. Introduction

Cancer is one of the leading causes of death, and despite the huge efforts of health-care systems and extensive research into novel diagnostic strategies and chemotherapeutic agents, cancer incidence and mortality are still growing [1]. Resistance to anticancer drugs represents a major obstacle to efficacious cancer therapies, and is considered to be responsible for 90% of cancer-related mortality [2]. The drug-dependent resistance mechanisms are often associated with ADME systems (drug absorption, distribution, metabolism, excretion) denoting a significant issue in clinical practice [3]. Genetic polymorphisms of drug-metabolizing enzymes, uptake and efflux transporters resulting in altered function are crucial in interindividual variations in drug exposure [4,5]. Most anticancer agents are enzymatically transformed into inactive, active and/or toxic metabolites; therefore, the function of drug-metabolizing enzymes (e.g., cytochrome P450 enzymes, CYPs) is of high importance in drug resistance [6,7]. One must also note the relevance of the germline polymorphisms of drug-metabolizing CYP enzymes to the therapeutic outcome of anticancer drugs, as an example of innate resistance. Prodrug type agents (e.g., tamoxifen, cyclophosphamide), requiring functional CYPs for the formation of active metabolites, have been demonstrated to lose their effectiveness in patients with loss-of-function *CYP* variants [8]. Besides the pharmacokinetic variability in the patients’ systemic circulation of a drug due to pharmacogenetic polymorphisms, the intratumoural drug disposition can also contribute to resistance to anticancer agents [9,10]. The complex nature of pharmacokinetic drug resistance is notably associated with (1) high variability of extratumoural and intratumoural drug concentrations, (2) synergistic interplay between the key systems involved in tumour-specific elimination (drug-metabolizing enzymes, transporters), and (3) strong genomic instability in cancer cells leading to a highly variable activity of these enzymes/proteins and even to resistant cell clones.

Increased efflux transport of anticancer agents mediated by ABC transporters (ATP-binding cassette transporters) has been demonstrated to decrease intratumoural drug accumulation and has been well established to be a critical reason for drug resistance in many tumours [11]. However, only a few studies have focused on the biotransformation of these drugs in tumour cells as one of the resistance mechanisms [12,13]. The anticancer drug-metabolizing capacity of tumour cells is a potential either for local activation of prodrugs to an efficient metabolite or for inactivation of anticancer agents in the tumour leading to resistance to the therapy. The expression and activity of drug-metabolizing enzymes in tumours has been assumed to influence the success of drug therapy and may be a prognostic marker for treatment response [14]. Copy number alterations of these enzymes and consequent increase (or decrease) in enzyme activities may be manifested in tumours due to genomic instability and alterations in chromatin structure. The most commonly used methods for the identification of gene copy numbers in cancer cells are next-generation sequencing and single-nucleotide polymorphism (SNP) arrays. Next-generation sequencing can detect large-scale and small-scale genomic changes from the high-throughput whole genome and whole exome sequencing data of cancer cells. Relatively high costs, huge space requirement for data storage, complex data analysis using specific bioinformatic tool(s) by a trained bioinformatician, and the inaccessibility of the generated data to the public contribute to the disadvantages of sequencing methods [15,16]. From SNP arrays, diverse information can be extracted, e.g., they can detect small-scale copy number gains and losses. The process of evaluation by trained personnel requires several platform-specific tools; thus, interpretation of the results may take a long time [17]. Alternatively, quantitative polymerase chain reaction (qPCR)-based methods can be applied for the identification of copy number alterations. These methods require relatively small amounts of sample DNA and are simple, open-format tools that allow the application of researcher-designed assays. Furthermore, simple evaluation of the results facilitated their widespread use [18]. In normal tissues, the PCR-based approaches generally assess the copy number of target genes by quantification relative to reference diploid genes (intrasample normalization), whereas in tumour tissues, the copy numbers are compared to those in normal tissues (intersample normalization). In the present study, the copy number alterations of two drug-metabolizing *CYP* genes, *CYP2C8* and *CYP3A4*, were determined in lung adenocarcinoma samples using normal lung tissues or blood samples for normalization.

Lung cancer is the second most common cancer in incidence, estimated to be 22.3% with 19.6% mortality in both sexes in Europe [1]. The most common subtype is adenocarcinoma with a 5-year survival rate of 15% primarily due to resistance to anti-tumour treatments [19]. Chemotherapeutic treatment typically includes taxanes, such as paclitaxel, often combined with platinum-based compounds. Paclitaxel is an antiproliferative agent that efficiently stabilizes microtubules inhibiting spindle function and the progression of mitosis, eventually triggering apoptosis [20]. CYP-mediated metabolism of paclitaxel includes CYP2C8 as the major catalyst forming the 6α-hydroxy-paclitaxel metabolite, while CYP3A4 is responsible for the minor 3′*p*-hydroxylation pathway [21]. These metabolites have no or a negligible effect on microtubule stabilization, which means that biotransformation results in inactivation of paclitaxel [22]. The genetic variability of CYP2C8 and CYP3A enzymes in patients may lead to variations in the exposure to paclitaxel and in patients’ response to the therapy. In vitro studies have revealed that SNPs in *CYP2C8*3* (2130G>A, rs11572080 and 30411A>G, rs10509681) and in *CYP2C8*4* (11041C>G, rs1058930) result in decreased enzyme activities [23,24], whereas *CYP3A4* and *CYP3A5* polymorphisms (*CYP3A4*1B*: −392A>G, rs2740574; *CYP3A4*22*: 15389C>T, rs35599367; *CYP3A5*3*: 6986A>G, rs776746) may also contribute to hepatic CYP3A activity and may have an impact on paclitaxel clearance [9]. Several studies have investigated the effects of patients’ *CYP2C8*, *CYP3A4* and *CYP3A5* genotypes on paclitaxel pharmacokinetics [25,26,27,28]; however, the direct involvement of CYP enzymes in therapy resistance has not been shown.

According to the hypothesis of the present study, altered copy numbers of paclitaxel-metabolizing CYP enzymes as an intrinsic genetic profile of tumour subclones may potentiate them for an increase in CYP expression and metabolic activity that may contribute to the emergence of clones resistant to paclitaxel therapy. These resistant cells are often present in the primary, non-treated tumour, leading to disease relapse and an unfavourable therapeutic outcome. The major aim of the present study was to develop a high-throughput TaqMan-based qPCR method for copy number analysis which is easily accessible and may provide an alternative to sequencing methods in everyday clinical practice. This approach may also facilitate better understanding of the therapy-resistance mechanisms emerging locally in the tumour.

## 2. Results

Alterations in copy numbers of paclitaxel-metabolizing CYPs (CYP2C8, CYP3A4) were aimed to be determined in primary lung adenocarcinoma samples prior to therapy. High-throughput qPCR methods were developed for the identification of copy numbers of the target (*CYP*) and the reference genes for which the primary lung adenocarcinoma, lung and liver metastasis samples of the OK18 patient were applied. Whole genome sequencing was used to confirm the copy numbers in the primary adenocarcinoma of the OK18 determined by qPCR-based methods. We attempted an investigation into the association between the copy number alterations of paclitaxel-metabolizing CYPs in the primary lung adenocarcinoma samples and therapeutic outcome in 17 patients under paclitaxel therapy. OK18 patient was excluded from the association analysis because his therapeutic regimen did not contain paclitaxel.

### 2.1. Copy Number Alterations in Tumour Determined by qPCR-Based Method

To improve the accuracy and reliability of the qPCR approach, we selected eight reference genes that are located on different chromosomes (chr1, chr4, chr6, chr7, chr14, chr15, chr17, chr19) and on various chromosomal regions (close to the centromere: *ALB*, *CD36*, *RPPH1*; close to the telomere: *TBP*; in between: *F5*, *B2M*, *MPO*, *BCKDHA*), and that are known to have two copies in a diploid genome (Appendix A). In tumour tissues, the copy number of each gene was calculated by the comparative ∆∆Ct method obtained from the tumour and the control tissue (blood or normal lung tissue) pairs of each patient, and the relative copy number of each gene was determined in comparison to the other nine genes in each sample (Figure 1a–c). The control tissues were expected to have a normal (diploid) copy number for both the reference and the target *CYP* genes; therefore, the relative copy numbers obtained by intrasample normalization were set to 1. In tumour samples, the acceptable range of the relative copy numbers where the copy numbers in the tumour and the reference control tissue were equal (intersample normalization), was within 0.8 and 1.25 (with a 90% confidence interval and α = 0.1). The outlier gene pairs with the relative copy number lower than 0.8 and higher than 1.25 were considered to indicate inequality, obtaining scores of −1 and +1, respectively. It should be mentioned that the relative copy numbers did not show discrete values in tumour samples, and the continuous values may be explained by intratumoural heterogeneity. When we compared the copy numbers of each gene to each other in tumour samples, the sum of copy number alteration scores assigned gene deletion (score ≤ −5) and multiplication (score ≥ 5) (Table 1).

For the development of qPCR-based copy number analysis, the primary lung adenocarcinoma as well as lung and liver metastases of a non-smoker patient (OK18) were analyzed (Figure 1, Appendix A). Relative copy number calculation indicated an increase in two genes, *F5* and *RPPH1*, and a decrease in *CYP2C8* in the primary tumour sample (the mean of the relative copy number for *F5*: 1.297, for *RPPH1*: 1.301 and for *CYP2C8*: 0.80), whereas in lung and liver metastases, both an increase and a decrease in relative copy numbers of several genes were detected (lung metastasis sample: *F5*: 1.514, *BCKDHA*: 1.681, *ALB*: 0.653, *CYP2C8*: 0.713; liver metastasis: *F5*: 2.289, *RPPH1*: 1.882, *BCKDHA*: 1.343, *CD36*: 1.295, *ALB*: 0.616, *CYP2C8*: 0.583) (Figure 1a–c). Furthermore, some alterations in copy numbers in liver metastasis were also observed in the primary lung tumour (multiplication of *F5* and *RPPH1*; deletion of *CYP2C8*) and in the lung metastasis (multiplication of *F5* and *BCKDHA*; deletion of *ALB* and *CYP2C8*) (Figure 1d). The sum of the scores for these genes was higher than 5 or lower than −5 (Table 1) except for *BCKDHA* and *CD36* in liver metastasis; therefore, in the particular tumour sample, these genes were considered to be inappropriate for normalization of the number of gene copies. For absolute quantification of gene copies in tumour samples, (1) the genes were applied as reference genes that were assessed to have the same copy number in the particular tumour as in the normal tissue (blood or normal lung tissue) with the sum of the scores within the range of 5 and −5, and (2) the average of the relative copy numbers was multiplied by 2 because of the diploidy of the control (normal) tissue. For the paclitaxel-metabolizing CYPs, deletion of *CYP2C8*, but not of *CYP3A4* was detected in primary lung tumour and in lung and liver metastases (the average copy number of *CYP2C8* in primary lung adenocarcinoma: 1.60, in lung metastasis: 1.47, in liver metastasis: 1.08). It also means that in a primary lung tumour, 40% of the cells and in lung metastasis, approximately half of the tumour cells displayed *CYP2C8* deletion, whereas in liver metastasis, on average one copy of *CYP2C8* was deleted from all the cells.

The copy number results assessed by the qPCR-based method were compared to those obtained by whole genome sequencing of the primary lung tumour of OK18 patient. We used ASCAT [29] to determine potential somatic copy number changes in the primary lung adenocarcinoma sample (Appendix A). This tool uses allele-specific piecewise constant fitting to model the copy number across the whole genome. The copy number profile of the sample was largely diploid and showed only low levels of heterogeneity. For each target and reference gene, the unrounded sums of the predicted copy numbers of the two alleles were taken as the local absolute ploidy level. The copy number alterations of the genes *F5*, *RPPH1* and *CYP2C8* identified by the qPCR-based method in the primary tumour sample were confirmed by next-generation sequencing, and the absolute copy numbers of both the target and the reference genes established by sequencing and qPCR-based methods were congruent with each other (for 10 genes, slope: 0.9498, r^2^ = 0.7524) (Figure 1e).

### 2.2. CYP Haplotype Distribution and Copy Number Alterations in Lung Adenocarcinoma Samples

For the estimation of patients’ paclitaxel-metabolizing capacity, *CYP2C8*, *CYP3A4* and *CYP3A5* alleles (*CYP2C8*3*, *CYP2C8*4*, *CYP3A4*1B*, *CYP3A4*22*, *CYP3A5*3*) most common in Caucasian populations were identified in lung adenocarcinoma patients on paclitaxel therapy (N = 17) (Table 2). Those who did not carry any of the *CYP2C8*, *CYP3A4* or *CYP3A5* polymorphisms were considered to have the *CYP2C8*1*, *CYP3A4*1* or *CYP3A5*1* wild-type alleles.

Most of the patients (N = 14) carried the *CYP2C8*1/*1* genotype, possessing the potential for having the functional CYP2C8 enzyme, whereas three patients were heterozygous for *CYP2C8*3*, predicting reduced CYP2C8 activity. The loss-of-function *CYP3A5*3* allele was identified in all patients, generally having the homozygous *CYP3A5*3/*3* genotype, and only three patients carried the functional *CYP3A5*1* allele (*CYP3A5*1/*3*). The SNP in the *CYP3A4*1B* (rs2740574) allele has been reported to be in genetic linkage with *CYP3A5*1* [30]; therefore, it was not surprising that two of the *CYP3A5*1* carriers also carried the *CYP3A4*1B* allele. The relative frequencies of *CYP2C8*3*, *CYP3A5*3* and *CYP3A4*1B* were similar to the prevalence reported in Caucasian populations (Table 2) [5,31]; however, no patient with *CYP2C8*4* or with *CYP3A4*22* was identified.

The multiplication of the *CYP2C8*3* allele is not expected to increase paclitaxel-metabolizing activity of the tumour cells; therefore, the multiplication of the functional wild-type allele is reasonable to consider in a tumour. We have designed the sequences of primers and probes for copy number measurements that were able to distinguish the wild-type and polymorphic *CYP2C8*3* alleles. Only those tumour samples were considered to carry multiplication of *CYP2C8* in which the *CYP2C8*1* copy number was increased. The *CYP3A4*1B* allele produces functional CYP3A4 enzymes, and multiplication of *CYP3A4*1B* may lead to an increase in CYP3A4 function similarly to *CYP3A4*1*. Of the 17 lung adenocarcinoma samples, 7 displayed an increase in *CYP2C8* and/or *CYP3A4* genes (multiplication of *CYP2C8* in 1 sample, of *CYP3A4* in 3, and of both *CYP2C8* and *CYP3A4* in 3 samples), predicting elevated paclitaxel-metabolizing capacity of these tumours (OK18 was excluded because he was under non-paclitaxel therapy). A significant association was observed between the multiplication of paclitaxel-metabolizing CYPs and the outcome of paclitaxel therapy (responders: complete remission; non-responders: progressive disease/exit). In the primary tumour samples from non-responder patients (N = 9), copy number alterations in *CYP2C8* and/or *CYP3A4* were detected more frequently than in the samples from responders (N = 8) (non-responders 6/9 vs. responders 1/8, *p* = 0.0498, Fisher’s exact test) (Figure 2). However, for reliable association analysis, an increase in the size of the patient groups would be required.

## 3. Discussion

Resistance to anticancer agents is often associated with patients’ drug-metabolizing capacity, and germline polymorphisms of drug-metabolizing enzymes have been demonstrated to contribute to chemoresistance [3,8]. An increased inactivation rate of anticancer drugs (e.g., vincristine, erlotinib) is attributed to normal and ultra-rapid metabolizer phenotypes related to wild-type and gain-of-function *CYP* alleles, whereas poor metabolism linked to loss-of-function allele(s) clearly results in the low bioactivation rate of prodrugs, such as tamoxifen and cyclophosphamide. Significant interpatient variability in paclitaxel pharmacokinetics has also been assumed to be the source of substantial variations in treatment responses as well as in adverse effects [32,33]. Although in vitro studies have demonstrated the role of CYP2C8 and CYP3A enzymes in paclitaxel metabolism [21,23], the impact of *CYP* genetic polymorphisms (*CYP2C8*, *CYP3A4* and *CYP3A5*) on paclitaxel clearance appears to be controversial in patients. A lower paclitaxel clearance has been demonstrated in *CYP2C8*3* carrier patients with ovarian cancer than in non-carriers [34], and in contrast, increased paclitaxel-metabolizing activity was attributed to *CYP2C8*3* allele in breast cancer patients [35]. Furthermore, Hennigsson et al. and Marsh et al. have suggested that the substantial interindividual variability in paclitaxel pharmacokinetics can be explained by variant alleles of neither *CYP2C8* nor of *CYP3A4* and *CYP3A5* [25,27]. The limited patient group size of the present study did not allow us to evaluate the impact of *CYP* polymorphisms on patients’ response to paclitaxel therapy; however, it should be mentioned that the *CYP2C8*3* allele was identified both in responders and in non-responder groups (1 and 2, respectively), whereas all the three heterozygous patients with the *CYP3A5*1/*3* genotype were non-responders.

In addition to interpatient variability in the pharmacokinetics of anticancer agents, the biotransformation activity of cancer cells has also been raised as one of the relevant resistance mechanisms [7]. Altered expression (protein and mRNA) and activities of drug-metabolizing CYPs have been demonstrated in several tumour types compared to the normal non-tumourous tissues [9,36]; furthermore, as a consequence of altered drug metabolism, drug-tolerant cell clones have been assumed to be developed that can contribute to drug resistance [37,38]. Genomic instability, one of the main characteristics of tumour cells, is often manifested in structural variations, including copy number alterations (multiplication or deletion) of genes, e.g., drug-metabolizing *CYP* genes. Transcriptional changes owing to *CYP* copy number alterations can influence the ability of tumour cells to metabolize anticancer drugs, fostering the development of drug-tolerant cells within the tumour and progression of the disease. Thus, the multiplication or deletion of *CYP* genes may become a potential biomarker for tumours insensitive to a particular anticancer agent due to altered drug metabolism. There are various methods available for the detection of copy number alterations, using micro-array or next-generation sequencing techniques [39]; however, the development of simple and reliable alternative methods for routine clinical practice is still a requirement. A sensitive qPCR-based method with an optimal normalization procedure was presented here for the assessment of the copy numbers of paclitaxel-metabolizing CYP2C8 and CYP3A4 in lung adenocarcinoma samples. The identification of copy numbers of the target genes in a tumour is challenging because of the possibility of copy number alterations in the reference genes and because of the heterogeneity within the tumour [40]. The reference genes in normal diploid human cells are in two copies, whereas multiplication or deletion of the target genes may occur. Inherited (germline) copy number variations have been reported for *CYP2D6* and *CYP2C19*; however, such variations have not been described for other drug-metabolizing CYPs in normal tissues [41]. Because in cancer cells, structural changes can occur in nearly any gene due to genomic instability, multiple reference genes located on different chromosomes were applied. As a subsequent step, the copy numbers of the eight reference genes and the two target *CYP* genes in the tumour samples were compared to each other and to those in normal tissues (blood or normal lung tissue), and the relative copy numbers were calculated. The scoring system described in the present study simplified the identification of those genes that underwent copy number alterations in the tumour. Finally, only those genes were selected as reference genes that were not implicated in the disease.

Tumour heterogeneity is an important issue that complicates the analysis of copy number alterations. Due to the intratumoural heterogeneity, genetically diverse tumour-cell subpopulations may exist within a single tumour site as well as across various disease sites in a single patient [42]. As an unsurprising consequence, (1) the relative copy numbers of the genes in tumour samples displayed continuous values, (2) and the copy number profile of the primary lung tumour was observed to be distinct from those in lung or in liver metastases (in the OK18 patient). The intertumoural heterogeneity in patients with the same cancer type may also result in variations in copy number profiles. The qPCR-based copy number analysis, however, identified an increase in *F5* (chr 1q24.2) in all primary adenocarcinoma samples of the present study (N = 18); therefore, it was considered to be inappropriate as a reference gene. The arm-level copy number gain of chr 1q has been demonstrated in several cancer types, including lung adenocarcinoma, which explained the multiplication of the *F5* gene in the samples of the present study [43,44].

Involvement of uptake and efflux transporters in resistance mechanisms is well described in cancer cells [11]; however, drug-metabolizing CYP activities in tumours are rarely linked to the selection of drug-tolerant cell clones. In lung adenocarcinoma samples from patients, CYP3A4 mRNA and protein expression has been demonstrated to significantly exceed the expression in normal lung tissues [45]. Furthermore, CYP2C8 expression was also detected in many human lung cancer cell lines [36,46]. Resistance to paclitaxel has been assumed to be partly attributed to overexpression of paclitaxel-metabolizing CYP enzymes. Hofman et al. investigated the impact of CYP3A4 and CYP2C8 overexpression on the taxane (docetaxel, paclitaxel) sensitivity of HepG2 cells, and concluded that CYP3A4 was of paramount importance for docetaxel insensitivity, whereas overexpression of the CYP2C8 enzyme was found to be associated with increased viability of the overexpressed cells under paclitaxel exposure [12]. In addition, paclitaxel has been revealed to efficiently activate the human pregnane X receptor (PXR) and to induce CYP3A4 expression and activity [38]. Furthermore, the paclitaxel resistance of A549 cells (human lung adenocarcinoma cell line) has been reported to be associated with the PXR-mediated transcriptional induction of CYP2C8 enzyme and Pgp transporter protein (ABCB1 transporter) [46]. Increased expression of CYP2C8 and/or CYP3A4 and the high paclitaxel-metabolizing activity of tumour cells are also assumed to be developed as a consequence of the multiplication of *CYP* copy numbers. The present qPCR-based method successfully established *CYP* copy number alterations in lung adenocarcinoma samples, and the present study was the first that reasonably assumed an association between *CYP* multiplication in tumours and the disease outcome of paclitaxel-treated patients; namely, multiplication of *CYP2C8* and/or *CYP3A4* copy numbers was observed more frequently in non-responder patients (progression of disease/exit group) than in responders with complete remission of the disease. To establish a resistance mechanism related to *CYP* copy number alteration in the tumour would require a group size larger than the 17 patients presented here, which was the limitation of this study. However, it should be emphasized that the primary aim of the study was to develop a simple method for routine clinical testing of copy number alterations which may result in an increase in the expression of paclitaxel-metabolizing CYPs, and may contribute to the selection of resistant cells in the tumour.

A further question was raised that the copy number alteration may extend beyond the *CYP2C8* gene, and the alterations in tumour suppressor genes or oncogenes surrounding the *CYP2C8* may also contribute to disease evolution. The chromosomal location of the *CYP2C8* gene (chr 10q23.33) is close to tumour suppressor genes, such as *PTEN* (phosphatase and tensin homolog, chr 10q23.31), being involved in cell cycle regulation and *CPEB3* (cytoplasmic polyadenylation element binding protein 3, chr 10q23.32), the regulator of EGFR transcription [47]. The copy number alterations that can have an impact on tumour development and progression often comprise deletion rather than multiplication of tumour suppressor genes [48]. The extended multiplication of the chr 10q23.3 region may be expected to manifest as a clinically relevant increase in CYP2C8 expression, and the negative outcome of the disease (progression or exit) in the present study was assumed to be associated with *CYP2C8* multiplication and the increased paclitaxel-metabolizing capacity of tumour cells rather than the multiplication of tumour suppressor genes presumed on the basis of genomic location in *CYP2C8* surroundings.

In conclusion, drug-resistance mechanisms related to the metabolism of anticancer agents are assumed to be associated with the drug-metabolizing capacity of the patients and/or the tumour. Therefore, it is of particular importance to identify genetic polymorphisms of drug-metabolizing CYPs and to establish an association between loss-of-function or gain-of-function *CYP* alleles and therapeutic outcomes. In addition, the copy number alterations of *CYP*s in the tumour may also contribute to the selection of resistant tumour cells. A high-throughput qPCR-based method was successfully developed for the detection of copy number changes in the tumour, and the scoring system, which was first applied, improved the identification of those genes that underwent deletion or multiplication and consequently were not appropriate to become a reference. The present method appears to be an alternative to next-generation sequencing in routine clinical practice. In the treatment naïve lung adenocarcinoma samples, the multiplication of paclitaxel-metabolizing CYP2C8 and/or CYP3A4 was detected, and the *CYP* copy number increase was more prevalent in non-responder patients than in responders to paclitaxel therapy. Although further research is required to confirm the findings of the present study and to validate the clinical utility of this approach, the implication of altered *CYP* copy numbers may provide an appropriate biomarker of therapy resistance and prognosis for clinical outcomes.

## 4. Materials and Methods

### 4.1. Patients

Patients (N = 18) treated with lung adenocarcinoma at the National Korányi Institute of Pulmonology (Budapest, Hungary) were enrolled in the present study. Written informed consent was obtained from all participants to perform genetic analysis of the tumour and blood or non-tumourous lung samples. The patients belonged to the Caucasian population, and their demographic and clinical data were recorded (Table 3). The patients’ response to the paclitaxel-containing therapy (responders: complete or partial remission; non-responders: progressive disease/exit) was defined according to the Response Evaluation Criteria in Solid Tumors (RECIST 1.1) [49].

### 4.2. CYP Genotyping

The patients who were on paclitaxel therapy (N = 17) were genotyped for *CYP2C8*, *CYP3A4* and *CYP3A5* (for *CYP2C8*3*, *CYP2C8*4*, *CYP3A4*1B*, *CYP3A4*22* and *CYP3A5*3*). The patients’ blood samples were used to identify the SNPs in *CYP2C8* [2130G>A (rs11572080), 30411A>G (rs10509681), 11041C>G (rs1058930)], *CYP3A4* [−392A>G (rs2740574), 15389C>T (rs35599367)] and *CYP3A5* [6986A>G (rs776746)]. Genomic DNA was extracted from blood samples using a Quick-DNA™ Universal Kit (Zymo Research, Irvine, CA, USA) according to the manufacturer’s instructions for biological fluids and cells. Hydrolysis SNP analyses for *CYP2C8*3* (rs11572080 and rs10509681), *CYP2C8*4* (rs1058930), *CYP3A4*1B* (rs2740574), *CYP3A4*22* (rs35599367) and *CYP3A5*3* (rs776746) were performed by PCR with self-designed TaqMan assays (primer and probe sequences are in Appendix A). Real-time PCR was carried out with 20 ng of genomic DNA by using a Luminaris Color Probe qPCR Master Mix (Thermo Fisher Scientific, Waltham, MA, USA). Allelic discrimination was based on two TaqMan probes, specific to the wild-type and the variant alleles labelled with FAM and HEX fluorescent tags, respectively.

### 4.3. Copy Number Analyses Using High-Throughput qPCR

Patients’ blood or non-tumourous lung tissue and tumour samples were used to determine the relative copy numbers of *CYP2C8* and *CYP3A4*. Genomic DNA was extracted from non-tumourous lung tissues and tumour samples using the Quick-DNA™ Universal Kit (Zymo Research, Irvine, CA) according to the manufacturer’s instructions for solid tissues, whereas genomic DNA isolated from peripheral blood for *CYP* genotyping was also applied for copy number analyses. To establish CYP copy number alterations, the relative copy number approach was followed using the qPCR method. Briefly, the copy number assays applied a TaqMan method in a duplex PCR using a pair of primers for each gene and a VIC-labelled probe for *RPPH1* and a FAM-labelled probe for any of the target CYPs (*CYP2C8*, *CYP3A4*) or seven of the reference genes (*ALB*, *B2M*, *BCKDHA*, *F5*, *CD36*, *MPO*, *TBP*). Primers and probes were self-designed based on the reference sequences in the National Center for Biotechnology Information reference assembly, except for RPPH1 for which the assay was commercially available (TaqMan^®^ Copy Number Reference Assay, Cat. No: 4403326, Thermo Fisher Scientific) (Table 4). In blood samples or non-tumourous lung tissues for intrasample normalization, the qPCR method compared threshold cycles (Ct) of the target *CYP* genes and the reference genes that were known to be non-variable in the copy content (stable two copies in a diploid genome), to generate ΔCt values. The copy number assessment was based on the target sequences in *CYP2C8* or *CYP3A4* normalized to the sequences in eight reference genes (*ALB*, *B2M*, *BCKDHA*, *F5*, *CD36*, *MPO*, *TBP* and *RPPH1*) (Table 4). However, in the tumour an alteration in copy numbers of the reference genes may also occur; therefore, intersample normalization was required. Namely, the ∆∆Ct method comparing the copy numbers in the tumour to blood or non-tumourous lung tissue as the reference control tissues was applied for the assessment of relative copy numbers of both the target and the reference genes (for calculation of relative copy numbers in tumour samples, see Section 4.5 Data analysis).

A Biomark HD^TM^ (Fluidigm, San Francisco, CA, USA) high-throughput microfluidics qPCR system with Flex Six^TM^ integrated fluidics circuit chips was employed for copy number analyses. Specific target amplification as an initial step was performed according to the manufacturer’s instructions (TaqMan PreAmp Master Mix, Fluidigm). Thermal protocol for specific target amplification included an initial activation at 95 °C for 10 min followed by 14 cycles of denaturation at 95 °C for 15 s and annealing/extension at 60 °C for 4 min. The preamplification products were diluted to 1:5 with dilution buffer (Suspension Buffer, Fluidigm) and used as template samples in the qPCR. Assay mixtures for duplex reactions were prepared using Luminaris Probe qPCR Master Mix with ROX (Thermo Fisher Scientific, Waltham, MA, USA), primers and probes for *RPPH1* (labeled with VIC) and either *CYP2C8*, *CYP3A4*, *ALB*, *B2M*, *BCKHDA*, *CD36*, *F5*, *MPO* or *TBP* (labeled with FAM). Each assay and sample was transferred into the inlets of the primed chip and loaded with an integrated fluidics circuit controller (Juno, Fluidigm). After loading, the PCR was carried out as follows: 50 °C for 2 min, 95 °C for 10 min and 40 PCR cycles at 95 °C for 15 s and at 60 °C for 1 min.

### 4.4. Whole Genome Sequencing

For validation of the copy numbers obtained by the high-throughput qPCR method, whole genome sequencing was performed in the primary lung tumour sample and in the peripheral blood (corresponding normal tissue) of a lung adenocarcinoma patient (OK18) (alignment files: accession ID EGAS00001003416), and genome-wide ploidy levels were determined using ASCAT [29]. This tool uses allele frequency and coverage information for a large set of SNPs across the whole genome. The SNPs were taken from a list of common variants from the 1000 Genomes Project. Briefly, we calculated coverage and variant allele frequency values using alleleCount (https://github.com/cancerit/alleleCount, access on 28 July 2023) on a custom mutation list containing common human SNPs spaced evenly at minimal distances of 10kb. Genomic positions inside annotated centromeres in the GRCh38 reference or ENCODE blacklisted regions [50] were omitted, and coverages were corrected by local GC contents and replication timing values, which were obtained using the dedicated helper scripts bundled with ASCAT. Finally, we ran the ASCAT pipeline to infer regional ploidy levels by considering the average coverage levels and allele frequency distributions in the predicted regions with changed copy numbers. To account for tumour heterogeneity, we used the unrounded copy number predictions.

### 4.5. Data Analysis

In normal tissues (blood and non-tumourous lung tissues), the relative copy numbers of the genes were defined as 1 because of the two copies of both the target and the reference genes in a diploid genome. In tumour samples, the copy numbers of both the target *CYP* genes (N = 2) and the reference genes (N = 8) were evaluated relative to those in blood or non-tumourous lung samples by the ∆∆Ct method (intersample normalization). Briefly, in tumour samples, the relative copy numbers of any two of the genes were considered to be equal with those in normal tissues if the 90% confidence intervals for the ratio of gene 1 and gene 2 in the tumour lay within the 0.80–1.25 acceptance range of copy number relative to normal tissue. The relative copy number of each gene pair was calculated and was scored as 0 if it was within the 0.8–1.25 range, −1 if it was <0.8 and +1 if it was >1.25. Alteration in the copy number of a gene in the tumour was established if the sum of the scores was ≤5 (deletion) or ≥5 (multiplication). For the calculation of the absolute copy number of a gene in the tumour, (1) these genes were applied as reference genes that were proved to be in two copies, (2) and the relative copy numbers were multiplied by 2. Association between alterations in *CYP* copy numbers in lung adenocarcinoma samples and therapeutic outcome was evaluated by Fisher’s exact test. A *p* value < 0.05 was considered to be a statistically significant difference. 

## Figures and Tables

**Figure 1 ijms-24-13380-f001:**
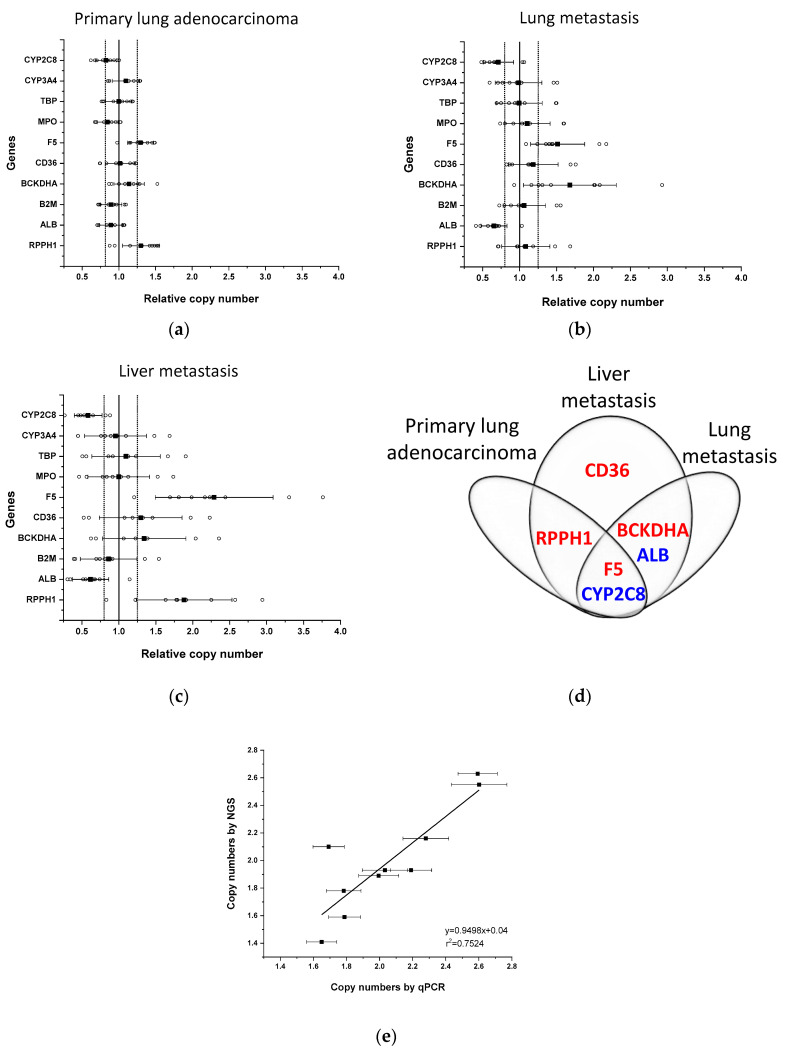
Copy numbers of genes in tumour samples of patient OK18. Relative copy numbers for each gene to those of the other nine genes (open circles) were assessed in the primary lung tumour (left inferior lobe, S6) (**a**) and in lung (right inferior lobe) (**b**) and liver metastases (right lobe) (**c**) by comparing to the normal lung tissue. Black squares represent the mean and whiskers are for standard deviation of relative copy numbers. Bold vertical line represents the normal relative copy number of the genes (set to 1). Dotted lines represent the acceptable range of normal copy numbers (0.8–1.25) in a sample. (**d**) Copy number alterations in genes in primary lung adenocarcinoma, liver and lung metastases (multiplication in red; deletion in blue). (**e**) Copy numbers of genes in primary lung adenocarcinoma determined by qPCR-based method and by next-generation sequencing. Whiskers represent the SD in copy numbers determined by qPCR-based method.

**Figure 2 ijms-24-13380-f002:**
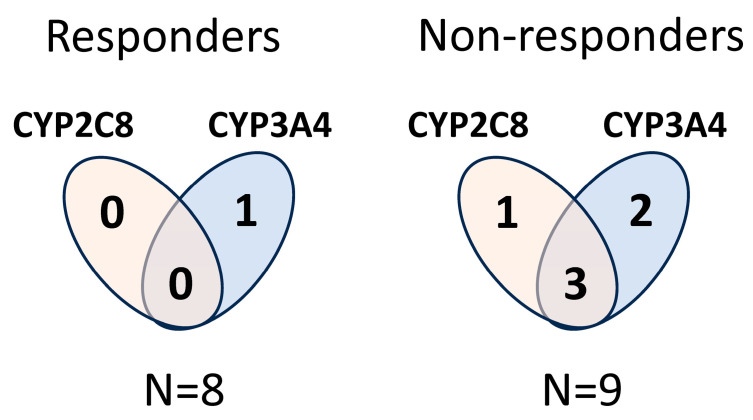
Copy number alterations in *CYP2C8* and/or *CYP3A4* in primary lung adenocarcinoma samples in responder and non-responder patients under paclitaxel therapy.

**Table 1 ijms-24-13380-t001:** The sum of the relative copy number scores for the genes in primary lung tumour and in lung and liver metastases of patient OK18.

Gene	Primary Lung Adenocarcinoma	Lung Metastasis	Liver Metastasis
*RPPH1*	**6 ***	0	**7**
*ALB*	−3	**−8**	**−8**
*B2M*	−3	0	−2
*BCKDHA*	3	**7**	3
*CD36*	−2	2	3
*F5*	**6**	**7**	**8**
*MPO*	−4	1	−1
*TBP*	−2	−1	0
*CYP3A4*	3	−1	−1
*CYP2C8*	**−5**	**−7**	**−7**

* Alterations (multiplication or deletion) in copy numbers are indicated in bold.

**Table 2 ijms-24-13380-t002:** *CYP* allele and genotype frequencies in lung adenocarcinoma patients (N = 17) on paclitaxel therapy and in Caucasian population.

*CYP* Allele/Genotype	N	Frequency (%)	
		Lung Adenocarcinoma Patients	Caucasian Population *
*CYP* allele			
*CYP2C8*			
**1*	31	91.2	79–90.5
**3*	3	8.8	6.5–14
**4*	0	0	3–7
*CYP3A4*			
**1*	32	94.1	83–95.5
**1B*	2	5.9	2–9
**22*	0	0	2.5–8
*CYP3A5*			
**1*	3	8.8	3–12
**3*	31	91.2	88–97
*CYP* genotype			
*CYP2C8*			
**1/*1*	14	82.3	81.8–85
**1/*3*	3	17.7	11.9–18.1
*CYP3A4*			
**1/*1*	15	88.2	90–90.9
**1/*1B*	2	11.8	7.7–9
*CYP3A5*			
**1/*1*	0	0	3.4
**1/*3*	2	11.8	11.6–19.71
**3/*3*	15	88.2	76.7–87.4

* *CYP* genotype frequencies in Caucasian population according to Gréen et al., Henningsson et al., Marsh et al. and Zanger et al. [5,25,27,28].

**Table 3 ijms-24-13380-t003:** Demographic and clinical data of lung adenocarcinoma patients.

		Complete Remission	Progressive Disease/Exit
Number of patients		8	10 ^b^
Gender (male/female)		2/6	4/6
Age (year) ^a^		64 (49; 74)	63.5 (22; 72)
Smoking	active	6	4
	quit after 1 year	0	2
	never	2	3
	NA ^c^	-	1
Primary tumour ^d^	1 (1a–1b)	-	2
	2 (2a–2b)	7	7
	3 (3a–3b)	1	-
Lymph nodes ^d^	0	3	5
	1	2	-
	2	2	3
	x	1	1
Metastasis ^d^	0	2	2
	x	6	7
Surgical intervention	right superior lobectomy	5	3
	left superior lobectomy	-	2
	right middle lobectomy	1	-
	right inferior lobectomy	2	1
	left inferior lobectomy	-	1
	atypical resection in right superior lobe	-	1
	atypical resection in left inferior lobe	-	1
	autopsy: left inferior lobe	-	1 ^e^
Therapy	paclitaxel+carboplatin	8	9
	gefitinib, cisplatin, afatinib	-	1 ^e^
	adjuvant radiotherapy	1	4

^a^ median (min; max); ^b^ one patient without TNM classification; ^c^ data not available; ^d^ TNM classification; ^e^ OK18 patient.

**Table 4 ijms-24-13380-t004:** List of genes and sequences of primers and probes for qPCR-based copy number measurement.

Gene	Gene Name	Chromosome Location	Forward Primer	Reverse Primer	Probe
*RPPH1*	Ribonuclease P RNA component H1	14q11.2	TaqMan^®^ Copy Number Reference Assay, Thermo Fisher Scientific Catalog number: 4403326		
*ALB*	albumin	4q13.3	TCT TCT GTC AAC CCC ACA	ATC TCG ACG AAA CAC ACC	GGC ACA ATG AAG TGG GTA ACC
*B2M*	beta-2-microglobulin	15q21.1	GAA TGA GTC CCA TCC CAT CT	CAG GGA AAC TAC TGG TTC AGA	CAG TAT CTC AGC AGG TGC CA
*BCKDHA*	branched-chain alpha-keto acid dehydrogenase	19q1.2	GTC GGC AGG TTA CCA CTA	GAA GGG CTG GGA CAT ACA	TAT CTG GGT GCC ATC TCC TC
*CD36*	cluster of differentiation 36	7q21.11	TGA AGC TGA AAT TGA AG TGG AG	AAC AGG TCT CAG TAC AGC AT	AGT CAG TTG TCC TCG TCG AAA TC
*F5*	coagulation factor V	1q24.2	TGA TCG AGG ATT TCA ACT CG	TCC CAT GAC AGA ACT CCT	AAG GTA AGA ACA CCC CCA CC
*MPO*	myelo-peroxidase	17p22.1	ACA CTA GGG TCA GTC CTT G	CAA GCC TGG GTC ATT ATG AC	CAT CTG GAG GAT TCC TGG AAC A
*TBP*	TATA box binding protein	6q27	CCT GCT CTG TTT TCA GAT GG	ATA GCT TTG CTT CCC TTT CC	AGT GGC ACT AAC GGT AAT TGT GT
*CYP2C8*	cytochrome P450 2C8	10q23.33	CGT TTC TCC CTC ACA ACC TTG C	ACT GTT AAG GTC AAT GAC GCA GA	CCT CAA TGC TCC TCT TCC CCA TCC CA
*CYP3A4*	cytochrome P450 3A4	7q22.1	CAG AGG TAG GTC TAA TTC AGT TCA	AGA TCA CCT TCT ATC ACA CTC C	ATC ACA CCC AGC GTA GGG C

## Data Availability

The CYP SNP data have been deposited in the European Variation Archive (EVA) at EMBL-EBI under accession number PRJEB64027 (https://www.ebi.ac.uk/eva/?eva-study=PRJEB64027, access on 28 July 2023), and the alignment file of the primary lung tumour and peripheral blood samples of OK18 patient are accessible in the European Genome–phenome Archive under study ID EGAS00001003416 (https://onlinelibrary.wiley.com/doi/epdf/10.1002/ijc.32159; access on 28 July 2023).

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
