# Peer review of "Potential Association of Cytochrome P450 Copy Number Alteration in Tumour with Chemotherapy Resistance in Lung Adenocarcinoma Patients"

_ijms, 2023, doi:10.3390/ijms241713380_

Round 1

Reviewer 1 Report

This research analyzed the association of CYP2C8 and CYP3A4 with the treatment response of lung adenocarcinoma in 17 samples. This research deals with drug metabolism by cytochromes P450.

I am not sure if the methods that are used are standard or they are in house methodology. In the results, the type of samples that was used, the number of cases and patients should be better described. I would be interesting to know if the results could be validated in other series as 17 samples is a small series.

Comments:

(1) Figure 2 shows the association of the copy number gain with non-responders. Due to the number of cases, the results is almost not significant. Although by PCR there is gain, the RNA levels or protein levels are unknown. Therefore, it is an speculation that a gain-of-function is present in that samples. Is this correct?

(2) Do copy number gain of these two markers associate with poor prognosis of lung cancer using other datasets?

(3) Regarding the CYP allele and genotype frequencies. How many cases were analyzed (17?). Why the number of cases (N) is not 17? Is the CYP trait governed by multiple alleles? Is it correct to use blood samples and do they represent the expression in liver?

(4) What is the role in the disease pathogenesis of each allele?

(5) In Figure it is shown the changes in patient OK18. How many samples were analyzed, and of what anatomical location?

(6) Regarding the NGS, what was exactly analyzed? Mutational landscape? All the exons and introns (whole genome sequencing).

Author Response

The authors thank the Reviewer for critical remarks that improved the manuscript. The point-by-point answers are in a separate file.

Reviewer 2 Report

The manuscript entitled "Potential association of cytochrome P450 copy number alteration in tumor with chemotherapy resistance in lung adenocarcinoma patients" is in my opinion very interesting because it identifies one of the various mechanisms of intrinsic chemoresistance in lung tumors, one of the most widespread and frequency of occurrence of resistance to chemotherapy. The results presented in this manuscript represent an excellent starting point for further studies in this field and for this reason they should be published.

Author Response

The authors thank the Reviewer for positive evaluation. We are to perform a prospective study in a cohort larger than the present one.

Round 2

Reviewer 1 Report

Thank you for the answers.

I do not have any additional comment.